# Inhibition of DUSP6 Activates Autophagy and Rescues the Retinal Pigment Epithelium in Sodium Iodate-Induced Retinal Degeneration Models In Vivo and In Vitro

**DOI:** 10.3390/biomedicines10010159

**Published:** 2022-01-12

**Authors:** Hao-Yu Tsai, Henkie Isahwan Ahmad Mulyadi Lai, Zhang-Yuan Chen, Tai-Chi Lin, Winnie Khor, Wen-Chuan Kuo, Jia-Pu Syu, Ping-Hsing Tsai, Yi-Ping Yang, Yueh Chien, Shih-Jen Chen, De-Kuang Hwang, Shih-Hwa Chiou, Shih-Jie Chou

**Affiliations:** 1Institute of Pharmacology, College of Medicine, National Yang Ming Chiao Tung University, Taipei 11217, Taiwan; ricky30284@gmail.com (H.-Y.T.); henkie.lai@gmail.com (H.I.A.M.L.); jeff19960114@gmail.com (Z.-Y.C.); jumperwinniebird@gmail.com (W.K.); figatsai@gmail.com (P.-H.T.); shchiou@vghtpe.gov.tw (S.-H.C.); 2Department of Medical Laboratory, Faculty of Health Sciences, University Selangor, Shah Alam 40000, Malaysia; 3Department of Ophthalmology, Taipei Veterans General Hospital, Taipei 11217, Taiwan; tclin6@vghtpe.gov.tw (T.-C.L.); sjchen@vghtpe.gov.tw (S.-J.C.); khuang@vghtpe.gov.tw (D.-K.H.); 4Department of Ophthalmology, School of Medicine, National Yang Ming Chiao Tung University, Taipei 11217, Taiwan; 5Institute of Biophotonics, National Yang Ming Chiao Tung University, Taipei 11217, Taiwan; wckuo@gm.ym.edu.tw (W.-C.K.); jpsyu@gm.ym.edu.tw (J.-P.S.); 6Department of Medical Research, Taipei Veteran General Hospital, Taipei 11217, Taiwan; molly0103@gmail.com (Y.-P.Y.); g39005005@gmail.com (Y.C.)

**Keywords:** DUSP6, ERK, NaIO_3_, autophagy, retinal pigment epithelium, autophagy flux, retinal degeneration

## Abstract

Autophagy plays a protective role in the retinal pigment epithelium (RPE) by eliminating damaged organelles in response to reactive oxygen species (ROS). Dual-specificity protein phosphatase 6 (DUSP6), which belongs to the DUSP subfamily, works as a negative-feedback regulator of the extracellular signal-regulated kinase (ERK) pathway. However, the complex interplay between DUSP6 and autophagy induced by ROS in RPE is yet to be investigated. To investigate the relationship between DUSP6 and autophagy, we exposed the ARPE-19 cell line and C57BL/6N mice to sodium iodate (NaIO_3_) as an oxidative stress inducer. Our data showed that the inhibition of DUSP6 activity promotes autophagy flux through the ERK pathway via the upregulation of immunoblotting expression in ARPE-19 cells. Live imaging showed a significant increase in autophagic flux activities, which suggested the restoration autophagy after treatment with the DUSP6 inhibitor. Furthermore, the mouse RPE layer exhibited an irregular structure and abnormal deposits following NaIO_3_ injection. The retina layer was recovered after being treated with DUSP6 inhibitor; this suggests that DUSP6 inhibitor can rescue retinal damage by restoring the mouse retina’s autophagy flux. This study suggests that the upregulation of DUSP6 can cause autophagy flux malfunctions in the RPE. The DUSP6 inhibitor can restore autophagy induction, which may serve as a potential therapeutic approach for retinal degeneration disease.

## 1. Introduction

Retinal degeneration is the leading cause of central vision loss among the elderly population [1,2]. A systematic literature review further indicated the significance of this condition from a global perspective and predicted its increasing prevalence due to the exponential ageing of the population [3]. Despite the fact that aging is the primary risk factor in the course of retinal deterioration, it is also influenced by other factors, including genetic susceptibility and oxidative stress. Retina has become a target of oxidative stress, since it is constantly exposed to the light stimuli needed for vision [4]. Oxidative stress occurs when the ROS level surpasses a certain threshold, which leads to the onset of a protective mechanism in the cells such as autophagy [5]. Furthermore, excess ROS disrupt autophagy homeostasis and damages the retina at the molecular level, thus diminishing its structural and functional integrity. Under normal circumstances, retinal pigment epithelial cells (RPEs) play a critical role in neutralizing the harmful radicals to maintain a redox homeostasis state in the retina [4]. Furthermore, RPEs phagocytize toxic lipid–protein aggregates derived from the outer layer of photoreceptors via autophagy [6]. However, this constant exposure to oxidative stress will eventually shift these RPEs to an exhausted stage, hindering their proper and regular functions [7]. Owing to the detrimental role of oxidative stress in the destruction of the retinal structure, many researchers have used oxidizing compounds (e.g., sodium iodate (NaIO_3_)) that can preferentially damage RPEs to generate in vitro and in vivo retinal degeneration disease models [8].

Autophagy is a catabolic process that removes unwanted or damaged cellular components via lysosomal digestion. Interestingly, several pieces of evidences have shown that autophagy is closely associated with oxidative stress [9,10]. Autophagy regulates cell biogenesis by recycling metabolic precursors and reduces oxidative stress by clearing toxic intracellular waste [11]. As mentioned, autophagy is crucial for cell homeostasis and in toxic lipid–protein aggregates metabolism in retinal physiology. However, autophagy dysregulation has been associated with increased susceptibility to oxidative stress, which may eventually lead to the progression of RPE degeneration in AMD [12,13]. Research has shown that sodium iodate can induce autophagy in RPE cells, leading to decreased mitochondrial activity by mitophagy, which in turn affects cellular viability [14]. The autophagic flux process was also found to be partly blocked in NaIO_3_-treated cells despite autophagy occurring [15]. Chan made the observation that the inhibition of autophagy in RPE cells enhances cell susceptibility to NaIO_3_, while the activation of autophagy is able to counteract the cell death mechanisms induced by NaIO_3_ [16].

The extracellular signal-related kinases (ERK1/2) are some of the main signaling pathways for the induction and maintenance of autophagy [17] and regulate the maturation of autophagic vacuoles [18]. To be specific, there are three subfamilies in the ERK1/2 family: serine/threonine phosphatase (PP2A, PP2C), tyrosine phosphatases (STEP, HePTP, PTP-SL), and dual-specificity protein phosphatases (DUSPs) [19]. Among the DUSPs family, DUSP6 specifically interacts with ERK1/2 and plays a counterbalanced role in the downstream regulation of ERK1/2 [20]. The inhibition of DUSP6 has been suggested to play a suppressive role in some diseases such as gastric or ovarian cancers [21]. Previous studies have also reported the upregulation of DUSP6 in response to external factors, including oxidative stress, and the participation of autophagy proteins (ATG) in regulation through the ERK1/2 pathway in many cell types [22].

Although the role of DUSP6 protein in ERK-mediated cell functions has been demonstrated in many cells, the relationship between DUSP6 and autophagy regulation in RPE cells under oxidative stress-induced cell death is still blurry. Thus, investigations on the mechanism regulating oxidative stress-induced autophagy in the RPE cells may offer new insights into retinal degeneration disease. In the present study, we investigated whether the regulation of DUSP6 could rescue retinal degeneration via the ERK1/2 autophagy pathway in both in vivo and in vitro retinal degeneration models through sodium iodate treatment. Our findings demonstrated that sodium iodate increases oxidative stress and autophagy in both models. We also studied the impact of a DUSP6 inhibitor, (E/Z)-BCI hydrochloride (BCI), on the regulation of the ERK1/2 pathway in retinal degeneration models. Upon BCI treatment, we observed a significant increase in the autophagic flux in vitro and structural repair in the retinal layer in vivo. Collectively, our results demonstrated that DUSP6 inhibition promotes autophagy flux activity through the upregulation of the ERK cascade.

## 2. Materials and Methods

### 2.1. Cell Line and Cell Culture

ARPE-19 cells were purchased from ATCC (ATCC^®^ CRL-2302™). Cells were cultured in Dulbecco’s modified Eagle’s medium (DMEM)/F-12 (Hyclone, Logan, UT, USA) supplemented with 10% FBS (Gibco, Grand Island, NY, USA), and 1% penicillin-Streptomycin (50 U/mL). Cells were passaged at 90% confluence and maintained at 37 °C and 5% CO_2_ with a 95% relative humidity.

### 2.2. Sodium Iodate and Protein Regulators Treatment in ARPE-19

Cells were seeded at a density of 2 × 10^6^ cells in 6 cm dishes for 24 h and treated with sodium iodate (NaIO_3_) (Sigma, Saint Louis, MO, USA) at concentrations 0, 2, 4, and 8 mM for 6 h at 37 °C. Cells were treated with the autophagy inhibitor Bafilomycin A1 (Sigma, Saint Louis, MO, USA) at 75 nM or DUSP6 inhibitor BCI (Sigma, Saint Louis, MO, USA) at 1.25 µM or 2.5 µM for 6 h at 37 °C. Proteins were collected after treatment for further study.

### 2.3. Immunoblotting Analysis

C57BL/6N mice were sacrificed and their eyes were collected. The excess muscle tissue outside the sclera was removed and then a needle was used to make a hole in the cornea then remove the whole cornea. Next, we eliminated the lens and retina, and finally lysed the remaining tissue RIPA buffer (10× Merck Millipore, Temecula, CA, USA) containing with 1% protease inhibitor. Meanwhile, in vitro ARPE-19 cells were collected and lysed with RIPA buffer containing with 1% protease inhibitor. Subsequently, cell lysates were centrifuged for 10 min at 4 °C and the supernatants were collected. Next, the protein concentrations were measured via the BCA Protein Quantification Kit. An equal weight of total protein was separated by electrophoresis on SDS/PAGE. After the proteins were transferred onto polyvinylidene difluoride (PVDF) membranes (Millipore, Bedford, MA, USA), the membranes were incubated with blocking buffer (1 × TBST and 5% skim milk) for 1 h at room temperature. After the blocking step, the membranes were incubated with the following primary antibodies at 4 °C overnight: DUSP6 (Abcam, Cambridge, UK), Phospho-ERK1/2 (Cell Signaling, Danvers, MA, USA), ERK1/2 (Cell Signaling, Danvers, MA, USA), LC3B (Novus, Centennial, CO, USA), SQSTM1/p62 (Abcam, Cambridge, UK), LAMP2 (Abcam, Cambridge, UK), Beclin-1 (Cell Signaling, Danvers, MA, USA), and β-actin (Sigma, Saint Louis, MO, USA). Next, membranes were washed three times with TBST and incubated with HRP-conjugated secondary antibodies including Goat anti-mouse IgG (Invitrogen, Carlsbad, CA, USA) and Goat anti-rabbit IgG (Invitrogen, Carlsbad, CA, USA) for 1 h at RT. The antigen-antibody complexes were detected by an enhanced chemiluminescence (ECL) substrate kit (Bioman, Millipore, Bedford, MA, USA). Bands were analyzed and quantified by the ImageJ (NIH, Bethesda, MD, USA) analysis software.

### 2.4. Autophagic Flux Assay

The cell autophagic flux was measured by the Autophagy Detection Kit (Abcam, Cambridge, UK). First, cells were pretreated with the autophagy inducer and chloroquine for 16 h as a positive control. After being treated with NaIO_3_, the cells were washed twice with 1 × assay buffer and 100 μL of Microscopy Dual Detection Reagent was added, which included green detection reagent and nuclear staining buffer. Samples were protected from light and incubated for 30 min at 37 °C. Next, the cells were carefully rinsed with 100 μL 1 × assay buffer and incubated for 20 min with 4% formaldehyde. After washing three times with 1 × assay buffer, the stained cells were analyzed by wide-field fluorescence microscopy.

### 2.5. Quantitative Real Time Polymerase Chain Reaction (qRT-PCR)

The cDNA was diluted to 200 ng/μL with sterile water and mixed with SYBR PCR Master Mix and primers. Amplification reaction was performed in a thermal cycler according to the manufacturer’s instructions. mRNA abundance was quantified using the threshold cycle method. The mRNA expression of GAPDH was used as an internal control for normalization.

### 2.6. 3(4,5. Dimethyl 2 Thiazolyl) 2,5 Diphenyl 2 H Tetrazolium Bromide (MTT) Assay

The MTT assay kit (Sigma, Saint Louis, MO, USA) was used for a cell viability test. ARPE-19 cells were plated at 5 × 10^4^ cells per well into 96-well plates for 24 h and treated with different concentrations of NaIO_3_ (0, 2, 4, 8, 16, 32 mM) for 6 h. Following treatment, the cells were washed in PBS and we added 10 μL of MTT working solution (5 mg/mL in phosphate buffer solution, Sigma, Saint Louis, MO, USA) for 4 h at 37 °C in a CO_2_ incubator. After incubation, 50 μL of DMSO (solubilizing reagent) was added to each well and then mixed by a micropipette. The presence of viable cells was visualized by the development of a purple color due to the formation of formazan crystals. Finally, the intensity was measured by the reading of OD540 on a microplate spectrophotometer (Tecan Austria GmbH 5082, Grobdig, Austria).

### 2.7. ROS Detection Assay (DHR-123 Assay)

Intracellular ROS was measured by the fluorescent probe dihydrorhodamine 123 (DHR-123) (Invitrogen, Carlsbad, CA, USA) to examine DHR-123 changes to rhodamine 123 (Rh-123) when oxidized by ROS. First, the ARPE-19 cells were plated at 5 × 10^5^ cells per well into 24-well plates for 24 h. Sodium iodate (NaIO_3_) was dissolved in Dimethyl Sulfoxide (DMSO). After treatment with different concentrations (0, 2, 4, 8 mM) of NaIO_3_ for 6 h, the cells were incubated with 2 μM DHR-123 dye for 30 min in the dark in 5% CO_2_ at 37 °C with a 95% relative humidity. Then, the fluorescence intensity of Rh-123 was measured by a fluorescence microscope (Olympus America, Melville, NY, USA) at a 488 nm excitation wavelength.

### 2.8. Animal

Healthy male C57BL/6N mice (8-week-old, about 20 g) were purchased from LASCO. The mice were housed under standard conditions of a 12:12 h dark–light cycle with access to standard rodent chow and water ad libitum. For each experimental group, 3 mice were used for the experiments. All the mice were treated according to the guidelines of the Association for Research in Vision and Ophthalmology (ARVO) Statement on Use of Animals in Ophthalmic and Vision Research, animal study was approved by Institutional Animal Care and Use Committee of Taipei Veteran General Hospital (IACUC), approval number: 2020-038 (1 January 2020).

### 2.9. Retinal Degeneration Model Established with Sodium Iodate Treatment by Intraperitoneal Injection

The sodium iodate solution (Sigma, Saint Louis, MO, USA) was diluted with PBS for AMD mimic model in vivo. The final concentration was 30 mg/kg for mice treatment and they were injected by intraperitoneal injection. The mice were then sacrificed and the mouse eyes were collected at different stages for further histological analyses. Following this established model, the protective effect of the DUSP inhibitor will be demonstrated through intravitreal injection and real-time image observation.

### 2.10. Histological Assessment of the Retina

Retina samples were fixed with 4% paraformaldehyde (Sigma, Saint Louis, MO, USA) overnight, followed by 1 × PBS. The methodology used for fixation, dehydration, clearing, infiltration, and embedding was developed according to that of Bio-Check Laboratories. Briefly, each retina sample was sectioned into 3 µm slices using vibratome (Leica, Buffalo Grove, IL, USA), followed by hematoxylin and eosin (H&E) staining. The morphology of the retina was observed using a light microscope (Olympus America, Melville, NY, USA). Next, the thicknesses of the retina, defined as the distance between the inner limiting membrane, Bruch’s membrane, and the outer nuclear layer (ONL), were measured by Image J (NIH, Bethesda, Maryland, USA).

### 2.11. Fundoscopy & Optical Coherence Tomography

After anesthesia with a mixed solution of Zoletil (50 mg/kg) and Ropum (10 mg/kg), a mouse retinal structure image was obtained with multi-contrast optical coherence tomography (OCT), according to previous reports [23]. The fundus photography images were acquired with a retinal imaging camera (Nikon, Shinagawa, Tokyo, Japan). Fundoscopy and OCT images were captured on the exact retinal space surrounding the optic nerve.

### 2.12. Immunofluorescence Staining

In vitro, the cells were seeded in 24-well plates at a density of 2 × 10^5^ per well and incubated at 37 °C with 5% CO_2_ for 24 h. Following treatment with 4 mM of NaIO_3_ for 6 h, the cells were washed twice with PBS and then fixed with 4% paraformaldehyde (PFA) (Sigma, Saint Louis, MO, USA) for 15 min. After rinsing three times with PBS, the cells were permeabilized and blocked with a blocking solution (0.3% BSA and 0.1% Triton X-100 in PBS) at room temperature for 1 h. After blocking, cells were incubated with primary antibodies overnight at 4 °C. The primary antibodies include DUSP6 (Abcam, Cambridge, UK), Phospho-ERK1/2 (Cell Signaling, Danvers, MA, USA), LC3B (Novus, Centennial, CO, USA), and SQSTM1/p62 (Abcam, Cambridge, UK). After incubation, cells were washed three times with PBST (0.1% Triton X-100 in PBS). Secondary antibodies (1:700) were applied for 3 h at room temperature. The secondary antibodies include IgG Alexa 488 goat anti-rabbit IgG and Alexa 594 goat anti-mouse IgG. Nuclei were stained with DAPI (1:1000) for 10 min at 4 °C and slides were observed by a light microscope (Olympus America, Melville, NY, USA).

In vivo, the sample slices were pretreated with bleaching solution (1% NaH_2_PO_4_ and 2.5% H_2_O_2_) overnight at room temperature. Next, the primary antibodies, including DUSP6, Phospho-ERK1/2, LC3B, SQSTM1/p62, and RPE65, were incubated with the sample slices overnight at 4 °C. After incubation, the samples were washed three times with PBST. Tests with secondary antibodies (1:700) were conducted for 1 h at room temperature. The secondary antibodies included IgG Alexa 488 goat anti-rabbit IgG and Alexa 594 goat anti-mouse IgG. After treating them with mounting solution, the slides were observed by a fluorescence microscope (Olympus America, Melville, NY, USA).

### 2.13. DUSP6 Inhibitor Treatment in Mice

Mice were divided into three groups: a control group, NaIO_3_ group, and BCI group. Mice in the BCI group were given NaIO_3_ at a dose of 30 mg/kg by intraperitoneal injection for 4 days and then given BCI (0.5 mg/kg) for 3 days through intravitreal injection. All the healthy controls received intra-peritoneal placebo injection and the NaIO_3_ group received the intra-vitreous placebo injection. The structure of the retina was analyzed by real-time images and H&E staining. Proteins were collected from retina for further study.

### 2.14. Statistical Analysis

The experiment data are presented as means ± SEMs. The normative distribution of the data was determined through the Kolmogorov–Smirnov test and non-parametric values were analyzed using the Mann–Whitney test. For statistical analysis, one-way analysis of variance (ANOVA) and Student’s *t*-test were performed; *p* < 0.05 was taken as significant, and highly significant differences in the statistics were accepted if *p* < 0.01.

## 3. Results

### 3.1. Sodium Iodate Disrupted the Autophagy Flux in ARPE-19 Cells

RPE malfunction is one of the major pathological characteristics of retinal degeneration, and autophagy is an important lysosomal degradation process that can remove damaged organelles and misfolded proteins in retinas. However, the mechanism of retinal degeneration remains unclear, and whether autophagy plays a role in it is still poorly understood. To investigate the pathological mechanism of retinal degeneration, ARPE-19 cells, which have been widely applied in studying retinal pathology, were exposed to sodium iodate (NaIO_3_), an oxidative induction reagent, to establish retinal degeneration models. To assess the expression of autophagic markers in RPE cells under the impact of NaIO_3_-induced stress, ARPE-19 cells were treated with different concentrations of NaIO_3_ (0, 2, 4, 8 mM) for 6 h (Figure 1a). We observed that NaIO_3_ induced the generation of autophagic vesicles in a dose-dependent manner (Figure 1b). To further examine the expression levels of autophagic marker under NaIO_3_ treatment, the total proteins were collected and subjected to an immunoblotting analysis. As demonstrated in Figure 1c, autophagic markers including LC3B-II, Beclin-1, and LAMP2 were gradually upregulated in NaIO_3_-treated cells_._ Furthermore, an increase in LC3B-positive speckles was also observed in the NaIO_3_-treated group, indicating autophagosome generation (Figure 1d). These data indicate the impact of NaIO_3_ on inducing autophagy in RPE cells.

Based on the immunoblotting and immunofluorescence staining data, we found that the expression of the autophagic marker p62 (p62/SQSTM1) is increased after NaIO_3_ treatment. p62 protein, a well-known substrate of selective autophagy, interacts with LC3B, resulting in the degradation of ubiquitinated substrates followed by its degradation in the autophagosome. However, the accumulation of p62 has previously been shown to be associated with the deficiency of autophagic flux under oxidative stress [24]. To further check the autophagic flux in RPE cells, the autophagic inhibitor Bafilomycin A1 (Baf-A1) was used to evaluate the cell response in NaIO_3_-treated ARPE-19 cells. Baf-A1 treatment led to an increase in LC3B-II expression in a dose-dependent manner, indicating the early stages of autophagy (Figure 1e). Meanwhile, the p62 expression levels remained unchanged despite the Baf-A1 treatment. Furthermore, NaIO_3_ treatment at any given dose did not affect the p62 mRNA level compared to the control ARPE-19 cells (Figure 1f). These results indicated that the elevated p62 expression induced by NaIO_3_ is not due to the disruption of proteasome activities or the upregulation of the p62 gene. Therefore, NaIO_3_ may have an effect on blocking late-stage autophagic flux in ARPE-19 cells.

### 3.2. NaIO_3_ Treatment Induces DUSP6 and MAPK Upregulation

In the model of retinal degeneration induced by the exposure of ARPE-19 cells to oxidized low-density lipoprotein (oxLDL), DUSP6 was found to be elevated in response to oxLDL treatment [25]. However, the treatment effect of NaIO_3_ on the expression of DUSP6 and the role of DUSP6 in autophagic flux in retinal degeneration remain unclear. Hence, we sought to investigate DUSP6 expression and its role in autophagic flux in RPE cells under NaIO_3_ treatment. An immunoblotting assay was conducted to examine the expression of DUSP6 and modulated kinase (p-ERK). The results indicated that DUSP6 expression was upregulated by NaIO_3_ treatment in a dose-dependent manner (Figure 2a). In addition, the p-ERK/ERK ratio, which represents the ERK activity, was also increased (Figure 2b). These data suggest that the DUSP6 and ERK pathways were stimulated by NaIO_3_ treatment in ARPE-19 cells. Collectively, these data indicate that DUSP6 expression was induced by the increased ERK activity in ARPE-19 cells under NaIO_3_ treatment.

### 3.3. BCI Promotes the Autophagic Flux via ERK Pathway Activity in ARPE-19 Cells

To investigate whether the inhibition of DUSP6 activity could regulate the autophagic flux against oxidative stress, we incubated ARPE-19 cells with NaIO_3_ in the presence or absence of BCI, the DUSP6 inhibitor, for a period of 6 h. Next, we analyzed the autophagic flux through an immunoblotting assay and immunocytochemistry analysis. The p-ERK/ERK ratio, which represents the ERK activity, was significantly increased in NaIO_3_-treated cells after BCI treatment in a dose-dependent manner (Figure 3a). Consistently, immunofluorescence staining showed that the speckles of p-ERK were increased after BCI treatment, despite the presence or absence of NaIO_3_ (Figure 3b). This result indicated that BCI stimulates p-ERK signaling under both normal and NaIO_3_- treated conditions.

Moreover, the late stage of autophagic flux was examined using immunoblotting and immunocytochemistry staining. The immunoblotting showed the upregulation of autophagic markers upon BCI treatment in a dose-dependent manner (Figure 3c). The ratio of LC3B-II/LC3B-I is shown in Figure 3d, while the quantification of p62 is displayed in Figure 3e. In addition, the fluorescence images reveal the aggregation of autophagosomes in the BCI-treated group. Moreover, the perinuclear p62 punctate, which represents intensive autophagic activity, was significantly upregulated in NaIO_3_ and BCI co-treated cells (Figure 3f). Furthermore, to estimate the autophagic flux in BCI-treated ARPE-19 cells, we introduced the GFP-LC3-RFP-LC3ΔG gene into ARPE-19 cells and monitored live images taken after BCI treatment. We observed that the number of GFP-positive puncta increased under NaIO_3_-induced oxidative stress, presenting a low autophagic flux. Interestingly, treatment with BCI led to the GFP-positive signal being shifted to an RFP-positive signal, indicating the induction of late-stage autophagy by BCI treatment (Figure 3g). Next, qPCR analysis showed that the mRNA level of p62 was upregulated in the BCI group, indicating that BCI could promote autophagic gene expression under oxidative stress in RPE cells (Appendix A). The positive effect of BCI in autophagic proteins signified that BCI effectively promoted autophagic flux through the blocking of DUSP6 activity to stimulate the p-ERK-mediated autophagy pathway. Taken together, these results suggest that the pharmacological activation of p-ERK by BCI could accelerate the autophagosome formation and the degradation of oxidative waste products in response to the oxidative stress induced by NaIO_3_.

### 3.4. DUSP6/p-ERK Axis Are Upregulated in NaIO_3_-Induced Retinal Degeneration In Vivo

NaIO_3_ has been widely used to induce RPE degeneration in vivo owing to its RPE-targeted oxidative damage [26]. However, the pathogenesis of RPE degeneration under the NaIO_3_-induced oxidative model is still unclear. To investigate the relationship between autophagy regulation and oxidative stress in vivo, we treated wild-type C57BL/6 mice with NaIO_3_ via an intravenous injection, followed by carrying out a retinal analysis of ultrahigh-resolution multi-contrast optical coherence tomography (OCT) [23] images and post-mortem histopathological sections for 1 week (Figure 4a). Abnormal deposits of migratory cells in the RPE layer (red arrows) were observed 4 days post-injection, and these escalated within 7 days (Figure 4a). The retina became thinner with the indistinct boundaries in different layers. In addition, the boundaries between the outer nuclear layer (ONL) and inner nuclear layer (INL) almost disappeared (Figure 4b).

The effects of NaIO_3_ on the retinal structure were further assessed by the hematoxylin and eosin (H&E) staining of retina tissue. In the control mice treated with saline, the retina showed the well-organized and clear boundary of retinal layers, including the ganglion cell layer (GCL), INL, and ONL. The retinal pigment epithelium (RPE) layer was smooth with even pigmentation, implying that the cells were in a healthy condition (Figure 4c). One day after injection with NaIO_3_, the retinal layers still remained regimented and showed a slightly melanin thicker morphology (Figure 4c). It is worth noting that the arrangement of ONL was significantly disorganized 4 days after the NaIO_3_ injection. In addition, the uneven pigmentation in the RPE layer indicated the degeneration of RPE (Figure 4c). Seven days after injection, the inner segment/outer segment junction showed severe disruption above the RPE layer. Meanwhile, the migration of RPE cells was clearly observed with the severe melanin accretion (Figure 4c). Furthermore, the retina became notably thinner due to the severe degeneration of photoreceptors and the RPE layer (Figure 4c). Taken together, this histopathological study demonstrated that the NaIO_3_ injection disrupted the retinal layer arrangement and induced severe pigmentation, accompanied with the accumulation of migratory cells and cell debris. Tight junction marker ZO-1 was used to validate the cell–cell junctions before and after treatment with NaIO_3_. Our results have shown that the cell–cell junction was damaged after cells were treated with NaIO_3_ compared to the control group (Appendix A). The histological examination of ocular specimens was performed, and we found that after treatment with NaIO_3_ the RPE layer had swelling and there had been a migration of pigmented cells into the OS layer. Based on previous studies, we suggest that the aggregation of cells was caused by the proliferation of microglials, and our data showed the Iba1 expression between the ONL and RPE layers (Appendix A) [27,28]. These findings successfully established a retinal degeneration disease model.

To identify the role of DUSP6 in response to oxidative stress in this retinal degeneration model, we collected eyeballs from the NaIO_3_-treated mice at different time points and examined the DUSP6 protein expression in the retina after NaIO_3_ treatment. The immunoblotting assays showed the upregulation of DUSP6 1 and 7 days after NaIO_3_ treatment. The expression of phosphorylated ERK was also significantly upregulated, especially at 7 days post-injection (Figure 4d). The quantification DUSP6, ERK, and phosphorylated ERK expression is shown in Figure 4e,f. In brief, we demonstrated that the DUSP6/ERK axis was upregulated in the mouse retina under NaIO_3_-induced oxidative stress, and this axis may play a critical role in the regulatory mechanism of retinal degeneration.

### 3.5. The Expression of Autophagic Markers Are Increased in Retinal Degeneration Model

Furthermore, to validate the autophagic flux in the NaIO_3_-induced retinal degeneration model in vivo, immunoblotting and immunofluorescence staining were performed. Immunoblotting analysis showed that the autophagic markers LC3B-II, p62, and Beclin-1 were all upregulated after the course of NaIO_3_ treatment (Figure 5a). Quantitative data showed a significant increase in LC3B-II expression under NaIO_3_-induced oxidative stress (Figure 5b,c). This result indicated autophagosome accumulation, demonstrating the initial activation of autophagic flux. The upregulation of p62 expression further represented the characteristics of autophagic flux (Figure 5a,c). Furthermore, immunofluorescence staining showed an elevated expression of LC3B in different retinal layers of NaIO_3_-treated mice, while p62 upregulation was predominantly found in photoreceptors and RPE layers (Figure 5d). These findings showed that NaIO_3_ can efficiently boost autophagic flux in the retina, particularly in the RPE cells.

### 3.6. BCI Activated the Autophagic Flux via ERK Pathway Upregulation In Vivo

Based on our observations, the inhibition of DUSP6 activity using BCI could activate autophagic flux via the upregulation of 62 mRNA expression in vitro. To confirm the protein diversification and signal pattern in mouse retinas after BCI treatment under NaIO_3_ induction, retinal sections were analyzed by immunofluorescence staining. We found that the phosphorylated ERK speckles around the ONL and RPE layers were increased in oxidative conditions and dramatically enhanced after BCI treatment (Figure 6a). This result indicated that BCI upregulates the ERK activity in mouse retinas, especially in photoreceptors and RPE cells. Moreover, the excessive co-localization of LC3B and p62 between the ONL and RPE layers was observed in the BCI-treated group, indicating the higher autophagy activity in photoreceptor and RPE cells (Figure 6b). In brief, these findings indicated that, under oxidative stress, the BCI treatment induced phosphorylated ERK activity and autophagic flux in mouse retinas, especially in photoreceptor and RPE cells.

To further investigate the protective effect of BCI treatment in vivo against NaIO_3_-induced oxidative damage, we conducted a histological examination and showed that the ONL boundary was significantly disorganized after the melanin deposition (Figure 6c, red arrows). Migratory cells were observed in ONL in the NaIO_3_-treated group, indicating the cell–cell junction loss of RPE under oxidative stress (Figure 6c). In the BCI group, the ONL disorganization was alleviated by BCI treatment in the presence of NaIO_3_-induced oxidative stress. The melanin deposition on the retina was also significantly reduced compared to that in the retinas of mice treated with NaIO_3_ only (Figure 6c). Moreover, immunofluorescence staining revealed that BCI treatment restored the ONL thickness, which was decreased by the thinner ONL after the NaIO_3_ treatment. These results indicate the protective effect of BCI in retinal cells (Figure 6d,e). Collectively, our findings show that the inhibition of DUSP6 protects against retinal degeneration through promoting autophagic flux under NaIO_3_-induced oxidative stress.

## 4. Discussion

The irreversible progression of retinal degeneration involves both inherited and acquired traits and ultimately causes cell death within the RPE layer and might result in blindness. Age-related macular degeneration (AMD) is the most common cause of retinal degeneration and is characterized by either the degradation of RPE cells (dry AMD) or choroidal neovascularization (wet AMD) [29]. In addition, other factors, including oxidative stress, inflammation, and hypoxia, can also cause retinal degeneration [13]. The exposure of RPE cells to oxidative stress results in the death of RPE cells and recapitulates the pathogenesis of AMD as a disease model. However, the underlying molecular mechanisms contributing to the progression of this disease, such as ROS-mediated signaling pathways, mitochondrial function, and autophagy, are yet to be investigated. To answer this question, we established a retinal degeneration model using NaIO_3_; an oxidative inducer on the RPE cell line, ARPE-19; and a mouse model, C57BL/6N. The mouse model received a single intraperitoneal injection of 30 mg/kg of NaIO_3_. These results showed that NaIO_3_ treatment caused extensive damage to the RPE cells and disrupt the mouse retina structures, especially in the RPE layer.

Autophagy-related proteins play a key role in autophagy, as they are the most tightly regulated components of the pathway. Beclin-1 regulates autophagy initiation and the lapidated form of microtubule-associated protein light chain 3-II (LC3-II) [29]. The lapidated form of microtubule-associated protein light chain 3-II (LC3-II) has been shown to be a reliable marker for completed autophagosomes. In addition, LAMP2 plays a critical role during the formation of lysosomes. Our immunoblotting data demonstrate elevated autophagy at multiple levels. First, the increase in Beclin-1, LC3-II, and LAMP2 protein expression indicates an increase in early autophagic flux in RPE cells under oxidative stress. Second, the phosphorylation of ERK protein was shown to be higher in NaIO_3_-treated RPE cells than in control RPE cells, hinting that RPE undergoes autophagy impairment. Notably, we also showed that DUSP6 acts as a negative regulator of the ERK pathway by decreasing the autophagic response in NaIO_3_-treated RPE cells. In agreement with our studies, it has been reported that autophagy impairment in RPE cells results in aggregation-prone proteins, cell degeneration, and eventually cell death [30]. A further novel finding is that BCI’s inhibition of DUSP6 activity could promote autophagic flux and restore the retinal structure.

The inhibition of autophagic flux is typically associated with aggregates or inclusion bodies positive for ubiquitin and associated with large perinuclear aggregates [31]. The impaired autophagy process has also been demonstrated in other neurodegenerative diseases, including Alzheimer’s disease, Parkinson’s disease, and Huntington’s disease [32]. The protein p62, also called sequestosome (SQSTM1), a protein that is itself degraded by autophagy, may act as a link between ubiquitinated proteins and the autophagic machinery, thus facilitating their degradation in the lysosome. p62 accumulates when autophagy is inhibited, and when autophagy is induced p62 levels decrease, suggesting that p62 may be used as an autophagic flux reporter. Planned comparisons revealed that the accumulation of p62 was significantly decreased in mouse retinas treated with BCI, which suggested the restoration of autophagic flux in RPE cells.

The mitogen-activated protein kinase (MAPK) pathway translates signals from mitogens into signals that regulate transcription and affect cell proliferation, differentiation, and apoptosis through the activation of protein kinase cascades [33]. Extracellular signal-regulated protein kinase (ERK1/2) is a member of the MAPK family that plays a critical role in delivering extracellular signals to the nucleus, as well as regulating cell proliferation, cell differentiation, and the cell cycle [34]. Research indicates that ERK1/2 is closely related to visual cycle regulation and plays a critical role in the survival of RPE and photoreceptor cells [35]. One study suggested that the MAPK/ERK signal transduction pathways may be involved in the reduced activity of autophagy in RPE cells on the nitrite-modified extracellular matrix (ECM) [36]. In this study, we showed the regulation of NaIO_3_-induced autophagy in an ERK1/2-dependent manner. This pattern of results is consistent with the previous literature, and we found that autophagy is involved in the NaIO_3_-induced upregulation of stress proteins LC3B and p62 and the activation of the ERK pathway [36]. These data suggest that the ERK1/2 pathway activates autophagic flux, which contributes to a possible survival mechanism under the oxidative stress condition. Dual-specificity phosphatase 6 (DUSP6) acts as a negative regulator in ERK-mediated cell function via the dephosphorylation of serine/tyrosine residues [20]. Interestingly, DUSP6 is upregulated by ERK activation, which stimulates a negative feedback loop to prevent ERK hyperactivation [37]. It has been shown that DUSP6 exhibits dual functions through mediating physiological signaling pathways while also contributing to the generation of oxidative stress depending on the different cell types and environments involved. In agreement with these reports, we found that the NaIO_3_-mediated ERK/p-ERK pathway upregulates the DUSP6 expression level in RPE cells, and that the inhibition of DUSP6 could rescue the impaired autophagy. These findings represent the regulatory mechanism of the DUSP6/ERK axis in retinal cells in accordance with previous studies.

It has been reported that BCI promotes the anti-tumor effect of cisplatin in a patient-derived xenograft (PDX) model [21] and attenuates the inflammatory response in LPS-activated macrophages via ERK1/2 upregulation [38]. To further confirm the role of DUSP6 in ERK-mediated autophagy in retinal cells, BCI was used to inhibit the DUSP6 activity to promote autophagic flux. Interestingly, we found that the p-ERK/ERK ratio was significantly increased upon BCI treatment, accompanied by the upregulation of autophagic markers in a dose-dependent manner. Furthermore, BCI treatment resulted in an elevated amount of p62 punctate around the cell nucleus in RPE cells under oxidative stress. In addition, the p62 mRNA level was upregulated following BCI treatment compared to the cells only treated with NaIO_3_. In accordance with these findings, other autophagy markers, including LC3B, Beclin-1, and Lamp2, were also upregulated by BCI treatment in NaIO_3_-treated cells. We further confirmed these data through the live imaging of NaIO_3_-treated RPE cells in a retinal degeneration model. The data demonstrated a significant increase in LC3B-II upon BCI treatment, indicating the inhibition of autophagic flux by DUSP6 under oxidative stress conditions [39]. These findings suggest that BCI may counteract the hazardous effect of oxidative stress in RPE cells via ERK1/2-mediated autophagy upregulation, offering clues for developing novel interventional strategies during disease states by restoring autophagic flux and preventing RPE cell dysfunction induced by oxidative stress.

## 5. Conclusions

In summary, the results from this study demonstrated that NaIO_3_ is an oxidant that can activate the ERK1/2 pathway and that the regulation of DUSP6 leads to suspending autophagic flux in RPE cells. Treatment with the DUSP6 inhibitor BCI modulates the autophagy activity and provides novel therapeutic help to restore autophagic flux and prevent retinal degeneration induced by oxidative stress (Figure 7).

## Figures and Tables

**Figure 1 biomedicines-10-00159-f001:**
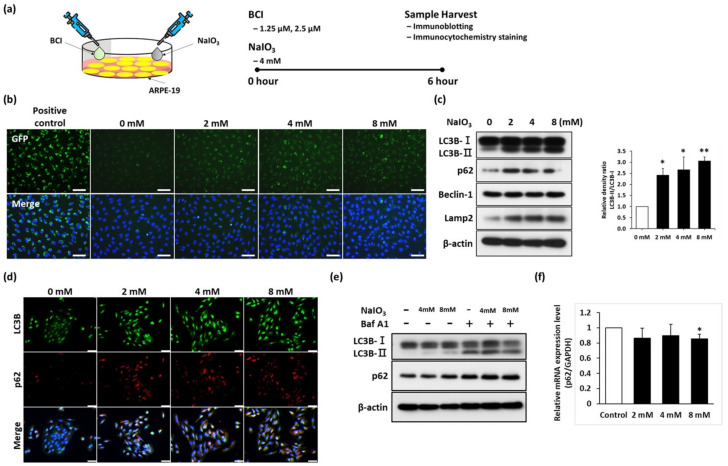
Sodium iodate disrupted the autophagy flux in ARPE-19 cells. (**a**) Schematic showing the protocol of in vitro NaIO_3_ treatment on ARPE-19 cells. (**b**) Autophagic vesicles in ARPE-19 cells with NaIO_3_ treatment were detected using a autophagic detection kit. Scale bar = 100 µm. (**c**) Representative blots analysis of autophagy relative protein expression after NaIO_3_ treatment. Quantitative analysis of immunoblotting. (**d**) Immunocytochemistry staining of LC3B and p62 after NaIO_3_ treatment in ARPE-19 cells. Nuclei were stained with DAPI. Scale bar = 50 µm. (**e**) Representative blots analysis of autophagic flux relative protein expression under Baf-A1 treatment. (**f**) Real-time quantitative PCR showed an mRNA level of p62; *n* = 3. Representative data from three independent experiments are shown. * *p* < 0.05 and ** *p* < 0.01.

**Figure 2 biomedicines-10-00159-f002:**
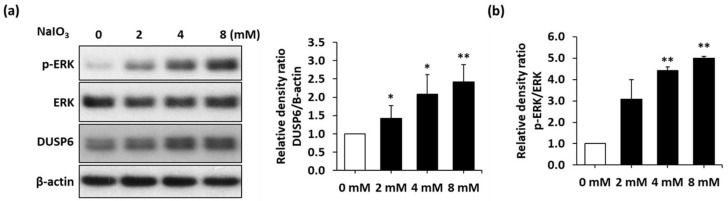
Upregulation of DUSP6 and MAPK after NaIO_3_ treatment. (**a**) Representative blots of DUSP6/p-ERK axis expression in ARPE-19 cells after NaIO_3_ treatment and the quantitative analysis of the DUSP6 expression level. (**b**) Quantitative analysis of the p-ERK level. Representative data from three independent experiments are shown. * *p* < 0.05 and ** *p* < 0.01.

**Figure 3 biomedicines-10-00159-f003:**
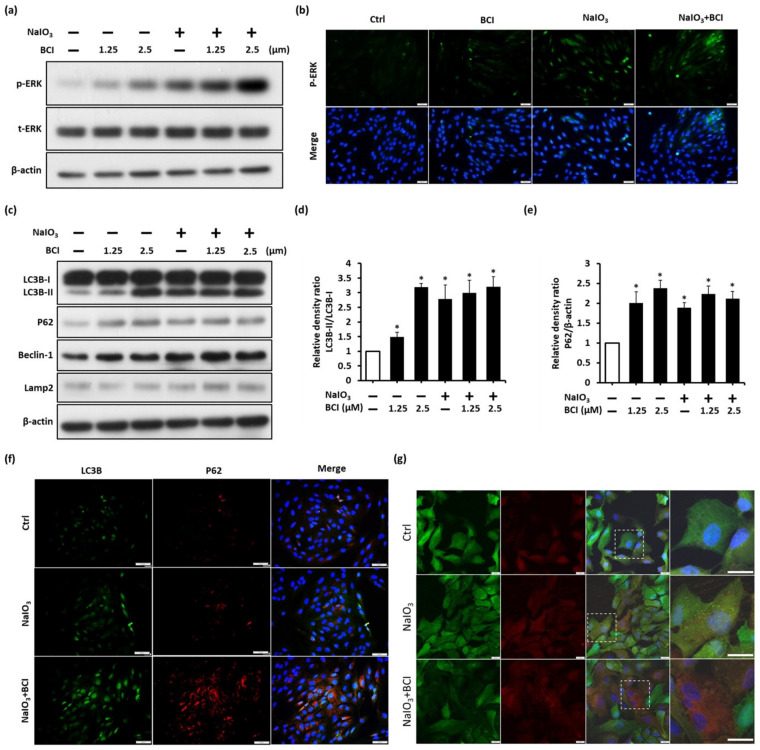
BCI promotes autophagic flux via ERK pathway activity in ARPE-19 cells. (**a**) Representative immunoblotting analysis of p-ERK/ERK axis expression in ARPE-19 cells treated with 1.25 and 2.5 µM of BCI in oxidative conditions. (**b**) Immunocytochemistry staining showed the p-ERK signals in ARPE-19 cells with or without BCI treatment in oxidative conditions. Scale bar = 50 µm. (**c**) Representative immunoblotting analysis showed the expression of autophagic relative proteins in AR-PE-19 cells treated with 1.25 and 2.5 µM BCI in oxidative conditions. (**d**,**e**) Quantitative analysis of immunoblotting. Representative data from three independent experiments are shown. * *p* < 0.05. (**f**) Immunocytochemistry staining exhibited the LC3B and p62 expression levels in ARPE-19 cells with BCI (1.25 µM) treatment in oxidative conditions. Scale bar = 50 µm. (**g**) Live-image of full-length LC3B (GFP) and cleavage form LC3B (RFP) expression level. Right subpanel represents the white boxed region of the image in the left subpanel with high magnification. Scale bar = 20 µm.

**Figure 4 biomedicines-10-00159-f004:**
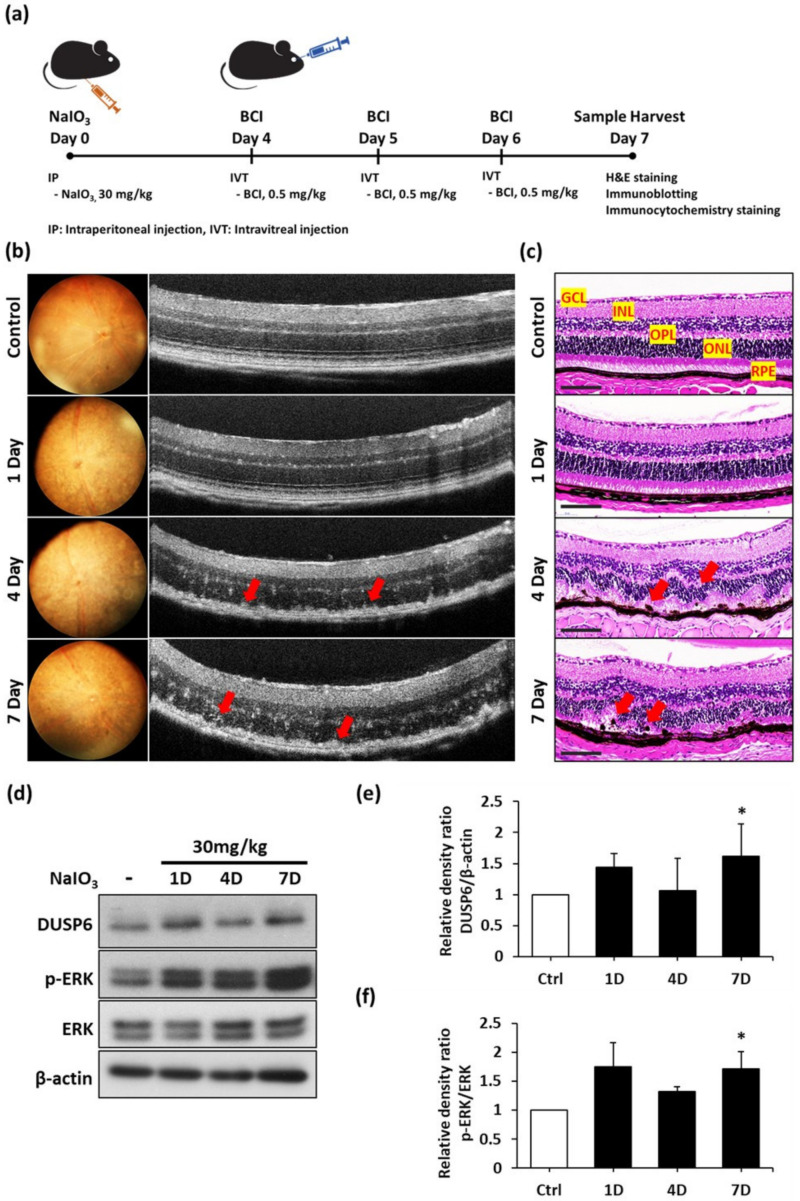
DUSP6/p-ERK axis was upregulated in NaIO_3_-induced retinal degeneration in vivo. (**a**) Schematic diagram illustrating the drug treatment and tissue preparation. (**b**) Real-time funds and retinal structures at different times post NaIO_3_ injection were observed by fundoscopy and OCT system. The red arrows indicate the irregulated structure on the RPE layer. Representative data from three independent experiments are shown. (**c**) H&E staining presenting the mice retinal structures after treatment with NaIO_3_. The abnormal deposits on the RPE layer is represented by the red arrows. Scale bar = 50 µm. (**d**) Representative blots showed the DUSP6/p-ERK axis expression level after NaIO_3_ injection in C57BL/6 mice. (**e**,**f**) A quantitative analysis is shown in (**d**). * *p* < 0.05.

**Figure 5 biomedicines-10-00159-f005:**
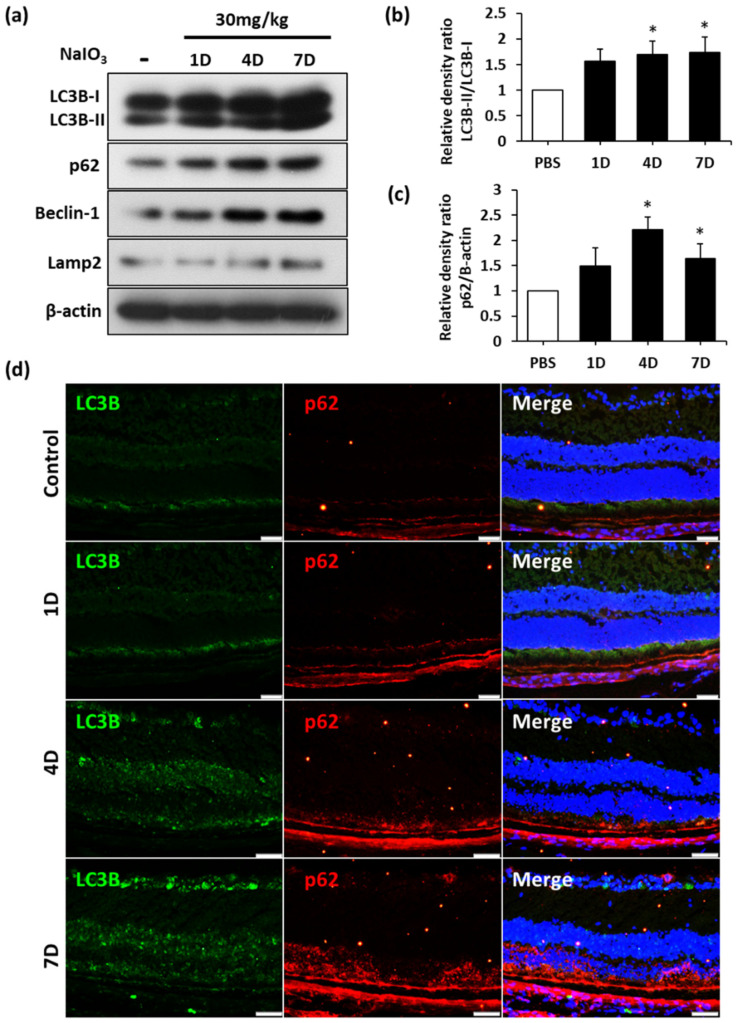
Autophagy relative genes are upregulated in NaIO_3_-induced retinal degeneration in vivo. (**a**) Representative blots showed autophagy relative markers after the NaIO_3_ injection of C57BL/6 mice. (**b**,**c**) Quantitative analysis of the Western blot. * *p* value < 0.05 in comparison to control. (**d**) LC3B and p62 expression levels in the retinas of mice given NaIO_3_ injections were observed via immunofluorescence staining. Scale bars = 50 µm.

**Figure 6 biomedicines-10-00159-f006:**
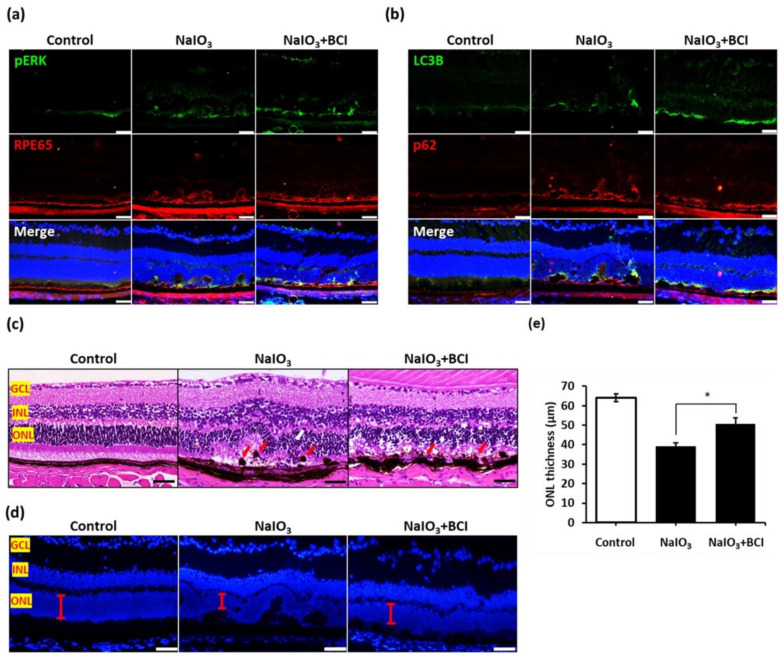
BCI activated autophagic flux via the upregulation of the ERK pathway in vivo. (**a**) Immunofluorescence staining of p-ERK in mice retinas with BCL treatment. 50 µm. (**b**) Immunofluorescence staining of LC3B and p62 in mice retinas with BCI treatment. 50 µm. (**c**) The morphology of mice retinas with BCI treatment was confirmed by H&E staining and (**d**) immunofluorescence staining. The red bar shows the thickness of the ONL layers. Scale bar = 50 µm. (**e**) Retinal thickness was calculated using Image J. * *p* value < 0.05 comparison to NaIO_3_ group.

**Figure 7 biomedicines-10-00159-f007:**
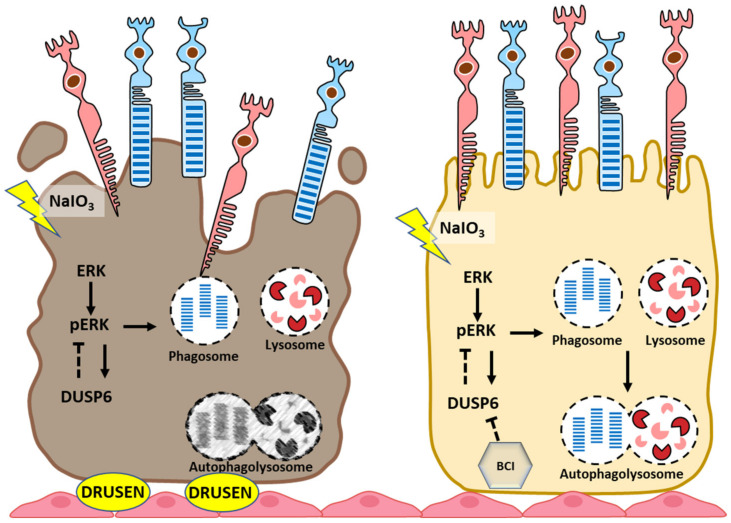
In conclusion, our data demonstrate that the inhibition of DUSP6 could protect RPE and the retina against NaIO_3_-induced oxidative stress-mediated autophagy dysfunction involving the ERK signaling pathway.

## Data Availability

Data contained within the article and the original data that support the findings of the present study are available from the corresponding author upon reasonable request.

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
