# Peer review of "Inhibition of DUSP6 Activates Autophagy and Rescues the Retinal Pigment Epithelium in Sodium Iodate-Induced Retinal Degeneration Models In Vivo and In Vitro"

_biomedicines, 2022, doi:10.3390/biomedicines10010159_

Round 1

Reviewer 1 Report

Comments and Suggestions for Authors

Dear authors,

The article titled ‘Inhibition of DUSP6 Activates Autophagy and Repairs the Retinal Pigment Epithelium in a Sodium Iodate-Induced Retinal Degeneration Models in vivo and in vitro’ submitted by Tsai et al to Biomedicines journal investigates the modulation of autophagy by oxidative stress and DUSP6 in the retinal pigment epithelium.

TITLE: The title properly pointed out the article's topic. However, the term “repairs” could be misleading because the study does not show whether BCI treatment induces a recovery or just preserve the surviving cells. In particular, from material and methods is not clear whether NaIO3 is removed from cells before BCI treatment. In the same way, it is not clear whether NaIO3 clearance precedes BCI treatments in mice. According to this information, it might be possible to recognize the repairing from the preserving property.

MINOR AND MAJOR CORRECTIONS NEEDED:

ABSTRACT: Abstract is comprehensive and clear. Essential information relevant to the findings of the study is incorporated. However, according to the observations above, authors should clarify about “recovering” and “repairing” processes (from line 26).

INTRODUCTION: The introduction is well written however, the state of art requires to be more extensively described in terms of metabolism and autophagy impairment due to NaIO3 exposure (e.g.: https://jbiomedsci.biomedcentral.com/articles/10.1186/s12929-019-0531-z; https://www.ncbi.nlm.nih.gov/pmc/articles/PMC6532221/; https://pubmed.ncbi.nlm.nih.gov/30020815/; https://iovs.arvojournals.org/article.aspx?articleid=2614664). Furthermore, it should be better underlined the step-forward with respect to previous studies (https://www.nature.com/articles/srep37279;). In addition, is not clear whether photoreceptors were directly affected to NaIO3 toxicity or they are secondary involved like a “bystander effect” (line 52) (https://www.ncbi.nlm.nih.gov/pmc/articles/PMC4049579/).

MATERIALS AND METHODS:

  1. I would suggest adding a scheme about the protocol of both, in-vitro and in-vivo treatments.
  2. The genetic background of C57BL/6N mice might spontaneously produce ocular abnormalities (10.1167/iovs.17-23513). Generally, C57BL/6J specie is more resistant to these pathological events. Please states whether healthy animals were screened before the experiments.
  3. Please update the immunoblotting analysis with description starting from mice tissues.
  4. Please states whether healthy control mice received the intra-peritoneal placebo injection and whether NaIO3 group received the intra-vitreous placebo injection.
  5. Please states whether statistic parametric tests used were preceded by normality test.

RESULTS: This section is straightforward and clear. The authors have explained the section based on the results obtained and categorized them. However, the following needs attention.

  1. In the legend of figures 1/2/3/4 statistical information are missing and their corresponding graphs lack asterisks. In figures 1/2/3/4/5, the control group in charts lacks the standard error bar.
  2. Like in figure 3, figure 6 should be updated with immunoblot analysis of pERK /LC3B/P62 and RPE65.
  3. Line 396. The presence of migratory cells might not be enough to confirm loss in the cell-cell junction. Diapedesis might occur without breaking occludence barrier, indeed. Therefore, specific immune-labeling of gap junctions, like ZO-1, (or some references about this event), is required.
  4. Line 336. The RPE migration is quite interesting however RPE detachment might be due to phagocytosis from migratory cells (10.1038/s41598-017-08702-7; https://iovs.arvojournals.org/article.aspx?articleid=2190442; https://iovs.arvojournals.org/article.aspx?articleid=2688228). Therefore, a microglia staining is required.

DISCUSSION and CONCLUSION: The discussion is quite clear but statistical tests in results are essential to confirm the interpretation of data. However, this study is mostly focused on characterizing autophagy in the NaIO3 –dependent impairment, and increasing data about DUSP6 mechanism might add weightage to the article and represent a much more significant step forward.

Reviewer 2 Report

The manuscript entitled “Inhibition of DUSP6 Activates Autophagy and Repairs the Retinal Pigment Epithelium in a Sodium Iodate-Induced Retinal Degeneration Models in vivo and in vitro” studies the relationship between the regulation of DUSP6 and retinal recovery via the ERK1/2 autophagy pathway in vitro (ARPE-19) cells and in vivo, (C57BL/6N mice) retinal degeneration models induced by sodium iodate treatment. The manuscript also covers the inhibition of DUSP6 by (E/Z)-BCI hydrochloride.

The manuscript is well written and easy to understand. The experiments carried out are adequate for their purpose of the investigation. The references used in the manuscript are recent and are adequate.  Regarding the novelty of the manuscript as far as I am concerned there are some works on the topic, but as the authors state this is the first time that the inhibition of DUSP6 is studied in the retinal pigment epithelium. I really liked the graphical conclusion; it summarizes the main findings of the manuscript in a clear way.

In my opinion the results shown in the present manuscript are interesting for a broader community.  Despite its great potential, the paper comes with a few minor issues which are addressed below:

  • Separate the degree from the number: 37°Cà37 °C.
  • Line 113 1hour à 1 hour.
  • Change statistical t and p value to italics.

Best regards

Round 2

Reviewer 1 Report

Dear authors,

Thank you for your answers. These are just some additional points for small improvements.

INTRODUCTION:

In addition, is not clear whether photoreceptors were directly affected to NaIO3 toxicity or they are secondary involved like a “bystander effect” (line 52)

(https://www.ncbi.nlm.nih.gov/pmc/articles/PMC4049579/).

Based on our observation, it seems that the result from a previous study

https://www.ncbi.nlm.nih.gov/pmc/articles/PMC4049579/ have similar results with our finding in Figure 4, which also mention by Wang et. al. that swelling and bundling of RPE cells lead to disorganization of photoreceptor and significant loss of RPE cells with thinning ONL. In addition, the study by Lin et al. also mentioned that without the support of the RPE, photoreceptor degeneration ensues (Lin et al. 2018). Hence, this study hypothesizes that sodium iodate affects RPE layers, and changes in the RPE layer indirectly affect photoreceptor layers.

I definitely agree with the authors and with all supporting references about that lack of the RPE functioning affect photoreceptors. However, whether contemporary there is a direct toxicity of NaIO3 on photoreceptors (maybe not immediately appreciable as showed by Hanus et al.) cannot be excluded as suggested by Wang et al.: “These results suggest that NaIO3 can directly alter photoreceptor function and survival.” Therefore, line 54 should be reviewed because the “RPE-selectivity” description of NaIO3 might be confused as “specific” instead of “preferential”.

MATERIALS AND METHODS:

  1. Please states whether healthy control mice received the intra-peritoneal placebo injection and whether NaIO3 group received the intra-vitreous placebo injection.

Thanks for the comments. We did perform intra-peritoneal placebo injection on the healthy control mice and NaIO3 group also performed the intra-vitreous placebo injection. Here, we included the NaIO3 group that received intravitreous injection. In this study, we used DMSO to dilute the BCI component, then as for the placebo injection, we use DMSO only to inject on the NaIO3 group. Figure below shown the placebo injection of NaIO3 and DMSO via intravitreous. From our observation, the NaIO3 group that received placebo injection share the similar condition as in NaIO3 treatment only.

Thank you for your description, could you please specify NaIO3 solvents in material and methods section (line 159) and could you also mention that you tested placebo injections? 

RESULTS: This section is straightforward and clear. The authors have explained the section based on the results obtained and categorized them. However, the following needs attention.

  1. Line 336. The RPE migration is quite interesting however RPE detachment

might be due to phagocytosis from migratory cells (10.1038/s41598-017-

08702-7; https://iovs.arvojournals.org/article.aspx?articleid=2190442;

https://iovs.arvojournals.org/article.aspx?articleid=2688228). Therefore, a

microglia staining is required.

We appreciate the reviewer's constructive suggestion and agree with the reviewers' comment. Indeed, we agree that RPE detachment might be due to phagocytosis from migratory of cells. Therefore, we have done an immunofluorescence staining for study. As shown in the figure below, we performed lba1 to show the microglia staining.

I thank the authors for their efforts. However, this panel lacks in the RPE labeling therefore it does not yet clarify whether the RPE actively migrates or it is detached by microglia phagocytosis. Corresponding images of pigmented epithelium acquired just via light-transmission might solve this issue. I would also suggest to add the panel to the manuscript, or just as supplementary images, and to review line 364 according to these up-dating.
